# A Mouse Model Suggests That Heart Failure and Its Common Comorbidity Sleep Fragmentation Have No Synergistic Impacts on the Gut Microbiome

**DOI:** 10.3390/microorganisms9030641

**Published:** 2021-03-19

**Authors:** Olfat Khannous-Lleiffe, Jesse R. Willis, Ester Saus, Ignacio Cabrera-Aguilera, Isaac Almendros, Ramon Farré, David Gozal, Nuria Farré, Toni Gabaldón

**Affiliations:** 1Barcelona Supercomputing Centre (BSC-CNS), 08034 Barcelona, Spain; olfat.khannous@bsc.es (O.K.-L.); jesse.willis@bsc.es (J.R.W.); ester.sausmartinez@bsc.es (E.S.); 2Institute for Research in Biomedicine (IRB Barcelona), The Barcelona Institute of Science and Technology, 08028 Barcelona, Spain; 3Unitat de Biofísica i Bioenginyeria, Facultat de Medicina i Ciències de la Salut, Universitat de Barcelona, 08036 Barcelona, Spain; ignaciocabrera.a@gmail.com (I.C.-A.); isaac.almendros@ub.edu (I.A.); rfarre@ub.edu (R.F.); 4Department of Human Movement Sciences, Faculty of Health Sciences, School of Kinesiology, Universidad de Talca, Talca 3460000, Chile; 5CIBER de Enfermedades Respiratorias, 28029 Madrid, Spain; 6Institut d’Investigacions Biomèdiques August Pi i Sunyer, 08036 Barcelona, Spain; 7Department of Child Health and Child Health Research Institute, The University of Missouri School of Medicine, Columbia, MO 65212, USA; gozald@health.missouri.edu; 8Heart Failure Unit, Department of Cardiology, Hospital del Mar (Parc de Salut Mar), 08003 Barcelona, Spain; 9Heart Diseases Biomedical Research Group, IMIM (Hospital del Mar Medical Research Institute), 08003 Barcelona, Spain; 10Department of Medicine, Universitat Autònoma de Barcelona, 08193 Barcelona, Spain; 11Catalan Institution for Research and Advanced Studies (ICREA), 08010 Barcelona, Spain

**Keywords:** metagenomics, microbiome, sleep fragmentation, heart failure, sleep apnea

## Abstract

Heart failure (HF) is a common condition associated with a high rate of hospitalizations and adverse outcomes. HF is characterized by impairments of either the cardiac ventricular filling, ejection of blood capacity or both. Sleep fragmentation (SF) involves a series of short sleep interruptions that lead to fatigue and contribute to cognitive impairments and dementia. Both conditions are known to be associated with increased inflammation and dysbiosis of the gut microbiota. In the present study, mice were distributed into four groups, and subjected for four weeks to either HF, SF, both HF and SF, or left unperturbed as controls. We used 16S metabarcoding to assess fecal microbiome composition before and after the experiments. Evidence for distinct alterations in several bacterial groups and an overall decrease in alpha diversity emerged in HF and SF treatment groups. Combined HF and SF conditions, however, showed no synergism, and observed changes were not always additive, suggesting preliminarily that some of the individual effects of either HF or SF cancel each other out when applied concomitantly.

## 1. Introduction

Heart failure (HF) is a prevalent disease associated with a poor, yet variable prognosis whose causal mechanisms are not entirely understood [1]. Comorbidities, such as sleep apnea, are frequent in patients with HF, and have been associated with a worsened prognosis [2]. The adverse outcomes associated with the co-existence of HF and sleep apnea have been attributed, at least in part, to excessive activation of the sympathetic autonomic nervous system [3,4], yet there is substantial variability underlying these relationships suggesting that other upstream factors may be also involved. Among these factors, the gut microbiome, a vast and complex polymicrobial community that coexists with the human host and is extraordinarily adaptable to a variety of intrinsic or extrinsic changes, plays an important role in the development of immunological phenotypes and in host metabolism [5], and could be implicated in the adverse outcomes of HF-sleep apnea [6].

Indeed, previous studies have shown evidence implicating the gut microbiome in the physiopathology and prognosis of HF [7]. HF is associated with reduced microbiome diversity [8] and a shift in the major bacterial phyla, resulting in a lower Firmicutes/Bacteroidetes ratio [9], and an increase in *Enterobacterales*, *Fusobacterium* and *Ruminococcus gnavus*, but also in a decrease in *Coriobacteriaceae*, *Erysipelotrichaceae*, *Ruminococcaceae*, and *Lachnospiraceae* [8]. Moreover, some intestinal microbial metabolites (e.g., trimethylamine-N-oxide (TMAO) and its precursors) are present in higher amounts in patients with chronic HF, and elevated levels of TMAO have been independently associated with an increased risk of mortality in acute and chronic HF [10]. Furthermore, patients with HF, present high blood levels of endotoxins, lipopolysaccharides (LPS), and tumor necrosis factor (TNF) [11] and have increased thickness of the intestinal wall, elevated intestinal permeability, and intestinal ischemia [11,12]. All these observations suggest a causal relationship between HF and gut dysbiosis and the edematous intestinal wall, epithelial dysfunction, and the translocation of LPS and endotoxins through the intestinal epithelial barrier promoting a mechanistic pathway that ultimately aggravates HF and leads to accelerated cardiac decompensation.

Sleep apnea is a highly prevalent comorbidity in HF [3], and is characterized by episodic hypoxia and intermittent arousals leading to sleep fragmentation (SF). Like many other disorders, sleep apnea has recently been associated with gut dysbiosis and systemic inflammation [13]. SF, one of the hallmark components of sleep apnea, has been less extensively examined than intermittent hypoxia [14,15], but studies to date have shown that it induces gut dysbiosis [16], and such changes are reflected by an increase in the Firmicutes/Bacteroidetes ratio, a preferential growth of the families *Lachnospiraceae* and *Ruminococcaceae*, and a decrease in *Lactobacillaceae* [16]. These changes are in turn associated with increased gut permeability, increased systemic LPS levels, and ultimately with systemic inflammation, which can further precipitate and maintain gut dysbiosis [17].

Given that both HF and SF are associated with gut dysbiosis and increased inflammation [17], we hypothesized that the coexistence of both conditions would result in a more marked alteration of the gut microbiome as compared with either condition in isolation. To test this hypothesis, we analyzed changes in the gut microbiome using a mouse model of HF and SF. However, contrary to our hypothesis, our results showed no additional effects of both conditions when applied together and suggested an attenuation of the changes that were observed in the conditions separately.

## 2. Materials and Methods

### 2.1. Animal Models Experiments

Forty male mice (C57BL/6J; 10 weeks old; 12 h light/dark cycle; water/food ad libitum) were randomly allocated into four groups (*n* = 10 each). In two groups, the mice were allowed to sleep normally: healthy control (C) and heart failure (HF). In two groups (SF, HF + SF), SF was imposed, and in two groups (HF, HF + SF) heart failure was induced. The animal experiment including the setting of the HF and SF models were approved by the institution ethical committee and has been recently described in detail [18].

HF was induced by continuous infusion of isoproterenol [18]. Briefly, mice were anesthetized by isoflurane inhalation and an osmotic minipump (model 1004, Alzet, Cupertino, CA, USA) was implanted subcutaneously in the flank. The pump delivered 30 mg/kg d^–1^ of isoproterenol (Sigma Aldrich. Munich, Germany; in sterile 0.9% NaCl solution) for 28 d. Buprenorphine (0.3 mg/kg, i.p.) was administered 10 min before surgery and after 24 h, and the suture was removed 7 d after surgery. Healthy animals were subjected to the same protocol with the only difference being that no isoproterenol was dissolved into the 0.9% NaCl pump medium. As described elsewhere [18], the effectiveness of the HF model in these animals was assessed by echocardiography after 28 d of isoproterenol infusion, confirming that mice in the HF groups had significant increases in left ventricular end-diastolic (LVEDD) and end-systolic(LVESD) diameter as well as significant reductions in left ventricular ejection fraction and fraction shortening.

Two days after surgery, SF was induced daily by means of a previously described and validated device for mice (Lafayette Instruments, Lafayette, IN), which is based on intermittent tactile stimulation with no human intervention. Sleep arousals were induced by a mechanical near-silent motor with a horizontal bar sweeping just above the cage floor from one side to the other side in the standard mouse laboratory cage. Each sweep was applied in 2-min intervals during the murine sleep period (8 a.m.–8 p.m.) for 28 d (until day 30 from surgery) [18].

At the end of the 4-week experiment (HF, SF, HF + SF and control), fecal samples were obtained directly from stool expulsion stimulated by manual handling of the animal and were immediately frozen at −80 °C and stored until further analysis. Two HF + SF mice died ending up with a final sample size for post samples of 38 (N_C_ = 10, N_HF_ = 10, N_SF_ = 10, N_HF + SF_ = 8)

### 2.2. DNA Extraction, Library Preparation and Sequencing

DNA was extracted from individual mice feces using the DNeasy PowerLyzer PowerSoil Kit (ref. QIA12855, Qiagen, Hilden, Germany) following the manufacturer’s instructions. After adding mice stool samples into the PowerBead Tubes (Qiagen), 750 μL of PowerBead Solution (Qiagen) and 60 μL of Solution C1 were added, and samples were vortexed briefly and incubated at 70 °C with shaking (700 rpm) for 10 min. The extraction tubes were then agitated twice in a 96-well plate using tissue lyser II (Qiagen) at 30 Hz/s for 5 min. Tubes were centrifuged at 10,000× *g* for 3 min and the supernatant was transferred to a clean tube. Furthermore, 250 μL of Solution C2 were added, and samples were vortexed for 5 s and incubated on ice for 10 min. After 1 min centrifugation at 10,000× *g*, the supernatant was transferred to a clean tube, 200 μL of Solution C3 were added, and samples were vortexed for 5 s and incubated on ice for 10 min again. Furthermore, 750 μL of the supernatant were transferred into a clean tube after 1 min centrifugation at 10,000× *g*. Then, 1200 μL of Solution C4 were added to the supernatant, samples were mixed by pipetting up and down, and 675 μL were loaded onto a spin column and centrifuged at 10,000× *g* for 1 min, discarding the flow through. This step was repeated three times until all samples had passed through the column. Furthermore, 500 μL of Solution C5 were added onto the column and samples were centrifuged at 10,000× *g* for 1 min, the flow through was discarded and one extra minute centrifugation at 10,000× *g* was done to dry the column. Finally, the column was placed into a new 2 mL tube to the final elution with 50 μL of Solution C6 and centrifugation at 10,000× *g* for 30 s.

Furthermore, 4 μL of each DNA sample were used to amplify the V3–V4 regions of the bacterial 16S ribosomal RNA gene, using the following universal primers in a limited cycle PCR:

V3-V4-Forward (5′-TCGTCGGCAGCGTCAGATGTGTATAAGAGACAGCCTACGGGNGGCWGCAG-3′) and V3-V4-Reverse (5′-GTCTCGTGGGCTCGGAGATGTGTATAAGAGACAGGACTACHVGGGTATCTAATCC-3′).

To prevent unbalanced base composition in further MiSeq sequencing, we shifted sequencing phases by adding various numbers of bases (from 0 to 3) as spacers to both forward and reverse primers (we used a total of 4 forward and 4 reverse primers). The PCR was performed in 10 μL volume reactions with 0.2 μM primer concentration and using the Kapa HiFi HotStart Ready Mix (Kapa Biosystems, Cape Town, South Africa). Cycling conditions were initial denaturation of 3 min at 95 °C followed by 20 cycles of 95 °C for 30 s, 55 °C for 30 s, and 72 °C for 30 s, ending with a final elongation step of 5 min at 72 °C.

After the first PCR step, water was added to a total volume of 50 μL and reactions were purified using AMPure XP beads (Beckman Coulter, Brea, CA, USA) with a 0.9× ratio according to manufacturer’s instructions. PCR products were eluted from the magnetic beads with 32 μL of Buffer EB (Qiagen) and 30 μL of the eluate were transferred to a fresh 96-well plate. The primers used in the first PCR contain overhangs allowing the addition of full-length Nextera adapters with barcodes for multiplex sequencing in a second PCR step, resulting in sequencing ready libraries. To this end, 5 μL of the first amplification were used as template for the second PCR with Nextera XT v2 adaptor primers in a final volume of 50 μL using the same PCR mix and thermal profile as for the first PCR but only 8 cycles. After the second PCR, 25 μL of the final product was used for purification and normalization with SequalPrep normalization kit (Invitrogen, Waltham, MA, USA), according to the manufacturer’s protocol. Libraries were eluted in 20 μL and pooled for sequencing.

Final pools were quantified by qPCR using Kapa library quantification kit for Illumina Platforms (Kapa Biosystems) on an ABI 7900HT real-time cycler (Applied Biosystems, Waltham, MA, USA). Sequencing was performed in Illumina (Sand Diego, CA, USA) MiSeq with 2 × 300 bp reads using v3 chemistry with a loading concentration of 18 pM. To increase the diversity of the sequences, 10% of PhIX control libraries were spiked in.

Two bacterial mock communities were obtained from the BEI Resources of the Human Microbiome Project (HM-276D and HM-277D), each containing genomic DNA of ribosomal operons from 20 bacterial species. Mock DNAs were amplified and sequenced in the same manner as all other murine stool samples. Negative controls of the DNA extraction and PCR amplification steps were also included in parallel, using the same conditions and reagents. These negative controls provided no visible band or quantifiable DNA amounts by bioanalyzer, whereas all of our samples provided clearly visible bands after 20 cycles.

### 2.3. Microbiome Analysis

The *dada2* pipeline (v. 1.10.1) [19] was used to obtain an ASV (amplicon sequence variants) table [20]. First, the sequence quality profiles of forward and reverse sequencing reads were examined using the *plotQualityProfile* function of *dada2*. Based on these profiles, low-quality sequencing reads were filtered out and the remaining reads were trimmed at positions 285 (forward) and 240 (reverse). The first 10 nucleotides corresponding to the adaptors were also trimmed, using the *filterAndTrim* function with the following parameters:“filterAndTrim(fnFs, filtFs, fnRs, filtRs, truncLen = c(285,240), maxN = 0, maxEE = c(10,10), truncQ = 1, rm.phix = TRUE, trimLeft = c(10,10), compress = TRUE, multithread = TRUE)”

Then, identical sequencing reads were combined into unique sequences to avoid redundant comparisons (dereplication), sample sequences were inferred (from a pre-calculated matrix of estimated learning error rates) and paired reads were merged to obtain full denoised sequences. From these, chimeric sequences were removed. Taxonomy was assigned to ASVs using the SILVA 16s rRNA database (v. 132) [21]. Next, a phylogenetic tree representing the taxa found in the sample dataset was reconstructed by using the phangorn (v. 2.5.5) [22] and Decipher R packages (v 2.10.2) [23]. We integrated the information from the ASV table, taxonomy table, phylogenetic tree and metadata (information relative to the samples such as the time, batch of the DNA extraction, and change of weight) to create a phyloseq (v. 1.26.1) object [24]. Positive and negative sequencing controls sequenced and included in the ASV table were removed from subsequent statistical analyses.

The metadata consisted of 11 variables: batchDNAextraction, sample, Time (indicating whether samples were taken prior to or post treatment); Box; SF.NORMAL.SLEEP (Sleep fragmentation or normal sleep); Animal; Pump (What substance was injected, Isoproterenol or Saline—control); Initial_weight; Final_weight; and Initial ecography (the value of which was “Ready” for all the animals). We created a new variable called Condition corresponding to the four different treatment groups: C, HF, SF and HF + SF.

Taxonomic composition metrics such as alpha-diversity (within-sample) and beta-diversity (between samples) were characterized. Using the estimate_richness function of the phyloseq package, we calculated the alpha diversity metrics including Observed.index, Chao1, Shannon, Simpson, and InvSimpson indices. Regarding the different beta-diversity metrics, we used the Phyloseq and Vegan (v. 2.5-6) packages to characterize nine distances based on differences in taxonomic composition of the samples including Jensen-Shannon Divergence (JSD), Weighted-Unifrac, Unweighted-unifrac, VAW-Gunifrac, a0-Gunifrac, a05_Gunifrac, Bray, Jaccard, and Canberra. We also computed Aitchison distance [25] using the cmultRepl and codaSeq.clr functions from the CodaSeq (v. 0.99.6) [26] and zCompositions (v.1.3.4) [27] packages.

Normalization was performed by transforming the data to relative abundances, and samples containing fewer than 950 reads were discarded and taxa that appeared in fewer than 5% of the samples at low abundances were filtered out:“prune_samples(sample_sums(object) ≥ 950, object)”“filter_taxa(object, function(x) sum(x > 0.001) > (0.05* length(x)), prune = TRUE)”

### 2.4. Statistical Analysis

Comparison of echocardiographic data between all groups at baseline was performed using one-way Analysis of variance (ANOVA). Comparison of echocardiographic data between all groups at day 30 was performed using two-way ANOVA followed by the Student-Newman–Keuls comparison method. The data is presented as mean ± SEM.

We used the Partitioning Around Medoids (PAM) algorithm [28], as implemented in the *cluster* library (v. 2.0.7-1), to explore clustering of the samples. We further evaluated this, performing a permutational multivariate analysis of variance (PERMANOVA) using the ten distance metrics mentioned above, and the *adonis* function from the *Vegan* R package (v. 2.5-6). The *Time* and *Box* variables were considered as covariates.

To identify taxonomic features (phylum, class, order, family, genus, and species) that show significantly different abundances among studied conditions, we used linear models, as implemented in the R package *lme4* (v. 1.1-21) [29]. Two different linear models were built: In the first one, the fixed effects were the *Condition* and *Time* variables and the random effects were the *batchDNAextraction* and the *Animal*, where this last one is an indicator of a paired analysis (tax_element ~ Condition + Time + (1| batchDNAextraction) + (1|Animal)). On the other hand, in the second linear model we included only post samples and, instead of the *Time* variable, we used as a fixed effect the *Change of weight* of the mouse models (*Final_weight-Initial_weight)*. In this case we only used as a random effect the batch (tax_element ~ Condition_POST_only + Change_of_weight + (1|batchDNAextraction)).

Analysis of variance (ANOVA) was applied to assess the significance for each of the fixed effects included in the models using the *Car* R package (v. 3.0-6) [30]. To assess particular differences between groups we performed multiple comparisons of the results obtained in the linear models using the *multcomp* R package by implementing the *glht* function and making use of the *Tukey* approach (v. 1.4-12) [31]. We applied *Bonferroni* as a multiple testing correction. Statistical significance was defined when *p* values were lower than 0.05 in all the analyses.

## 3. Results

### 3.1. Experimental Modelling of HF and SF

To test whether HF and SF showed synergistic effects we designed an experiment (see Materials and Methods) in which four groups (*n* = 10) of male mice (the gender where this condition is most common) were subjected for four weeks to each of the following conditions (see Materials and Methods):Heart failure (HF): This condition was induced by continuous infusion of isoproterenol, as previously described [18].Sleep fragmentation (SF): SF was induced daily by means of a previously validated device for mice (Lafayette Instruments, Lafayette, IN), based on automated intermittent tactile stimulation. Stimulation was applied in 2-min intervals during the murine sleep period (8 a.m.–8 p.m.), as described earlier [18].Combination of HF and SF (HF + SF) in which both conditions were induced in the same mice.Control: where no condition was induced. Before the start of the experiment and at the end of the 4-week experiment (HF, SF, HF + SF and control), fecal samples were obtained directly from stool expulsion and frozen at −80 °C until further analysis (Figure 1).

### 3.2. Characterization of the Microbiome

We used a 16S metabarcoding approach of the V3–V4 region and a computational pipeline (see Materials and Methods) to assess the microbiome composition before and after the treatment, in the different groups. The number of reads observed in each sample ranged from 25,053 to 121,981 with a mean of 58,030.99 (Rarefaction curve, Appendix A). Overall, we identified 128 and 114 different taxa at the genus and species levels, respectively. We classified 56.76% reads at the genus level, and the five most abundant genera were *Akkermansia, Alistipes, Bacteroides**, Lachnospiraceae_NK4A136_group* and an unclassified Muribaculaceae (F.Muribaculaceae.UCG).

We calculated different beta diversity metrics (see Materials and Methods, Section 2.3 Microbiome Analysis) and we produced multidimensional scaling (MDS) plots based on them such as the one represented in this section (Figure 2). We observed that sample stratification was significantly driven by *Time* (*p* < 0.05 Adonis, in all distance metrics except VAW_GUNifrac). This finding suggests that the microbiota of both treated and control mice had evolved significantly during the four weeks of the experiment (Figure 2a). In addition, we observed that samples clustered in two main enterotypes [32] (Figure 2b), which showed a significant relationship with the *Time* variable according to Bray–Curtis dissimilarity (Chi-square, *p* = 3.228 × 10^–6^).

### 3.3. Alpha Diversity

When considering all the samples together, the alpha diversity showed a tendency to increase at the end of the experiment (Figure 3a), although not significantly (*p* > 0.05, Wilcoxon). However, when comparing alpha diversity before and after the treatment within each group, the control group (C) but not the others, had a significant increase in alpha diversity (Figure 3c), whereas a trend toward a decrease in alpha diversity was noted for HF.

We also observed differences in alpha diversity between mice subjected to the different conditions. When considering only the samples after the experiment, we observed that both HF and SF groups had significantly lower alpha diversity, as compared to animals in C and (HF + SF) conditions (Figure 4a). When considering all samples, SF mice also showed a significantly lower alpha diversity as compared to the other groups (Figure 4b). This indicates the existence of differences in the basal microbiota before the start of the experiment and highlights the need to focus on changes occurring during the experiment rather than simply comparing final states.

### 3.4. Changes in Microbial Composition

We observed particular differences in abundance at different taxonomic levels according to the fixed effect variables used in the two different linear models: In the first linear model, all the samples were included (*n* = 78) and we studied the effect of both the *Condition* and *Time* variables, whereas in the second linear model we included only the samples after the experiment (*n* = 38), and focused on the Condition and Change of weight variables (Table 1).

For instance, according to the first linear model we obtained 47 differentially abundant taxa at the species level according to the *Time* variable. From these taxa, 11 were differentially abundant according to both the *Time* and Condition variables: *Bacteroides acidifaciens*, *Bifidobacterium* spp., F.Atopobiaceae.UCS, *Bacteroides* spp., *Rikenellaceae_RC9_gut_group* spp., F.Lachnospiraceae.UCS, *Ruminococcaceae_UCG.014* spp., *Ruminococcus* spp., *Allobaculum* spp., *Dubosiella* spp., and *Faecalibaculum* spp., whereas 15 and 36 taxa were separately reported for *Condition* and *Time*, respectively. (Appendix A).

We did not observe significant differences in both Bacteroides and Firmicutes phyla. However, when comparing their ratio between conditions we observed a tendency to decrease in all the groups but an increase, although not significant, when having both HF and SF conditions together (Appendix A).

On the other hand, applying the second linear model which only considered post-exposure samples, we observed 32 significantly differentially abundant species according to the Condition variable. Applying a multiple comparison test, the comparison with the most differences was C versus HF (Figure 5 and Appendix A). Note that we observed more changes when comparing HF and SF to healthy controls separately instead of when mice were exposed to both conditions. This supports the above-mentioned results, in which the alpha diversity was lower in HF or SF separately when compared to either C or HF + SF.

Six taxa at the species level were significantly altered by both the Condition and Change of weight variables: *Ileibacterium valens*, *Mucispirillum schaedleri*, F.Peptococcaceae.UCS, *Anaerotruncus* spp., *Ruminococcus* spp. and *Allobaculum* spp., while 26 taxa were only significantly differentially abundant according to the Condition variable (Table 2).

## 4. Discussion

In the present study we used a mouse model to assess the impact on the gut microbiome composition under conditions of HF and SF, and the combination of the two perturbations, which is frequently present in patients suffering from heart failure who go on to manifest sleep apnea. Overall, the study presents a clear separation between the samples before and after the induction of the conditions, including among the mice in the control group. This clustering may be produced by the anticipated evolution of the microbiome over time, a phenomenon that has been reported in several other studies of the mouse gut microbiome [33]. Interestingly, an increase in the abundance of the family Rikenellaceae, including the genus *Alistipes* (*p*-value = 1.86 × 10^–9^) in the post group samples (after four weeks of experiment) emerged, taxa that have previously been reported as being overrepresented in old mice and in elderly humans [34,35].

The overall alpha diversity was increased in the post-exposure samples, but this finding was only statistically significant in the control group. This suggests that species richness is significantly higher after the four weeks of the experiment when the mice are allowed to maintain their normal activities and are devoid of any of the experimental exposures, thereby corroborating earlier studies showing that older individuals exhibit more species overall than juveniles [36]. These results support the notion of an evolving gut microbiome during mouse development and underscore the importance of including samples taken at the start and at the end of the experiments to control for that variation. Importantly, the variation in species richness differed among the treated groups, wherein those exposed to only one of the relevant conditions displayed diminished species richness. Our findings concur with previous studies that showed an alteration in the microbiome in both HF and SF conditions and a decreased alpha diversity in HF patients [8,37].

The alteration of both Lachnospiraceae and Ruminococcaceae observed herein has also been noted by others in both isolated HF or SF models [8,16]. As mentioned, when applying a multiple comparison test considering only post samples, the largest differences were between C and HF. One example of a species that is altered is *Bacteroides acidifaciens*, which decreased in HF compared to C. *B. acidifaciens* has been linked to decreased obesity and to improved insulin sensitivity [38], is more abundant in individuals with high-fiber diets and acetate supplementation, and has been reported to play a role in the regulation of the circadian cycle in the heart [38,39]. Since a disturbance in the circadian cycle can cause cardiovascular complications [40,41], a decrease in *B. acidifaciens* may serve as an indicator of increased risk for deterioration of the underlying cardiac insufficiency. Interestingly, we also found this species to be decreased in SF samples compared to controls (*p*-value = 0.00025). This could also be due to the same reason, since a disturbed circadian cycle can lead to fragmented sleep, or alternatively, SF could induce the changes in gut microbiome that then disrupt the circadian cycle and elicit increased risk for cardiac decompensation in HF.

When we restrict our attention to the HF models, we observed an increase in the species *Ileibacterium valens* and the genera *Defluviitaleaceae_UCG.011*, *Ruminococcaceae_UCG.014*, *Ruminococcus, Allobaculum* and *Oxalobacter* compared to healthy controls. On the other hand, in addition to the mentioned increase of *B. acidifaciens*, we also observed a decrease in the species *Mucispirillum schaedleri* and the genera *Odoribacter, Alistipes, Mucispirillum, Lactococcus, Lachnoclostridium, Anaerotruncus, Oscillibacter, Dubosiella*, and *Anaeroplasma.* In previous studies, *Ruminococcaceae_UCG.014* abundance was found as significantly positively associated with serum trimethylamine N-oxide (TMAO) levels, which were associated with coronary atherosclerotic plaque and increased cardiovascular disease risk [42]. The genus *Ruminococcus* was also found increased in HF models [43], and was related to the inflammation that is observed in HF patients by the disruption of the gut barrier through either the translocation of gut bacterial DNA, endotoxins, or both, into the bloodstream [44]. It is known that both a high-fat diet (calorie-dense obesogenic) and aging cause inflammation in HF through an alteration of the microbiome such as increasing the phylum Firmicutes, specifically the genus *Allobaculum* [45], which in our study was found as significantly more abundant in HF than in C. Both *Alistipes* and *Oscillibacter* were also reported in previous studies as decreased in chronic HF patients [43].

Regarding the SF models, we observed increased *Muribaculum* and *Faecalibaculum* at the genus level, and decreased *B. acidifaciens* at the species level and *Lactococcus, Lachnoclostridium, Harryflintia*, and *Dubosiella* at the genus level. It is known that melatonin plays a beneficial role in the stabilization of the circadian rhythm [46] and a recent study reported that melatonin inhibits *Faecalibaculum* [46,47]. In our study we observed an increase of this genus. Therefore, this reduction can be an indicator of reduced melatonin bioavailability, and consequently reflects a destabilization of the circadian rhythm in SF-exposed mice. Our results also support past findings, whereby the genus *Lachnoclostridium* was reported as underrepresented in chronic intermittent hypoxia in guinea-pigs [48]. Hypoxia can be a consequence of a sleep disorder such as sleep apnea. We also found in the bibliography that *Harryflintia* was positively associated with a circadian clock gene (Cry1) whose mutations were related to sleep disorders [49].

When considering the coexistence of both HF and SF conditions compared with control mice, we detected only a very small number of differences, namely, an increase of *Muribaculum* and a decrease of *Bilophila.* Neither of these genera was previously related to these conditions. Overall, contrary to our initial hypothesis, our results show no strong synergism between the HF and SF conditions as their individual effects were not potentiated when applied in combination. Rather, the changes when the two conditions were combined were less apparent than when applying each condition individually, both in terms of changes in the alpha diversity and in the number of altered taxa. This suggests some level of antagonism between the two conditions, which may influence the microbiome in opposite directions, resulting in some of these effects cancelling each other out.

With this, we support recent studies suggesting that SF has no deleterious effect in the cardiac function [18] and surprisingly that chronic intermittent hypoxia increased the systolic function and improved left ventricular contractility in HF models, having a positive effect in the cardiac function [50,51]. These observations need to be further investigated as they may have implications for possible therapies.

One of the limitations of the study was that only male mice were considered. Although the use of females is warranted in all studies, we focused on male mice for two reasons. First, the majority of patients with HF, dilated ventricles, and reduced ejection fraction are males [52]. Second, having only one sex may control a possible confounding factor, so that the effect observed is more likely to be due to the conditions under study. However, some studies suggested that there are sex differences in the mechanisms that mediate sleep [53] and a better prognosis of HF in women [54]. Furthermore, sex differences have been found in relation to the gut microbiome. Thus, studies taking into consideration both sexes may be carried out in order to better elucidate the impact of both conditions in the gut microbiome.

Another two limitations of our study are the small sample size within each condition, limiting the statistical power of the findings, and the use of a mouse model, which entails some organism-dependent differences in the gut microbiome, such as distinct relative abundances in the dominant genera [55]. However, these are promising preliminary results with statistical significance adjusted by the sample size. Furthermore, mouse models are widely used standard models in experimental research that allow for the design of controlled experiments and for which the potential for gut microbiota research has previously been validated [55,56]. Certainly, our results could be validated in future studies by comparing gut microbiota profiles in a large cohort of patients affected by either HF and SF, or by the two comorbidities.

## 5. Conclusions

In summary, we have shown that the gut microbiome contains potential markers of heart failure and of sleep fragmentation when these conditions are evaluated separately. The inflammation observed in HF and SF could be mediated by alterations in abundance of particular taxa. Finally, when the two conditions were applied concomitantly, the alterations in the mouse gut microbiome were milder and virtually disappeared, suggesting some level of antagonism between the effects for HF and SF.

## Figures and Tables

**Figure 1 microorganisms-09-00641-f001:**
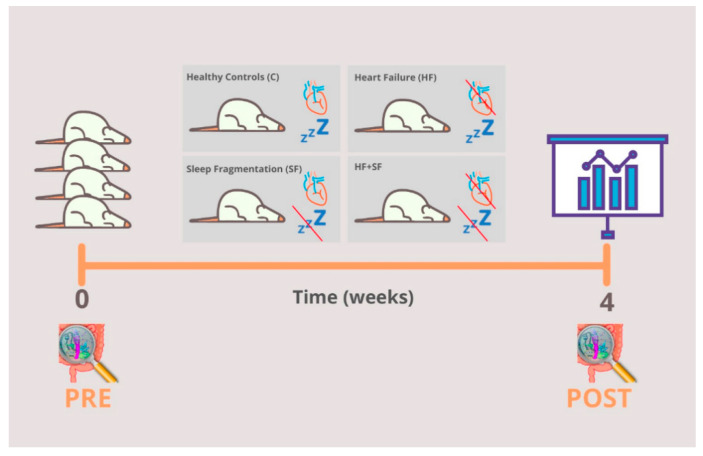
Experimental design of the study. Forty male mice (*n* = 40) randomly distributed in four groups: Healthy controls (C), heart failure (HF), sleep fragmentation (SF) and the combination of both conditions (HF + SF). The microbiome profiles of fecal samples obtained from these models were studied before and after a 4-week induction of the conditions. At the end of the experiment two HF + SF mice died, ending up with a final sample size of 38 (N_C_ = 10, N_HF_ = 10, N_SF_ = 10, N_HF + SF_ = 8).

**Figure 2 microorganisms-09-00641-f002:**
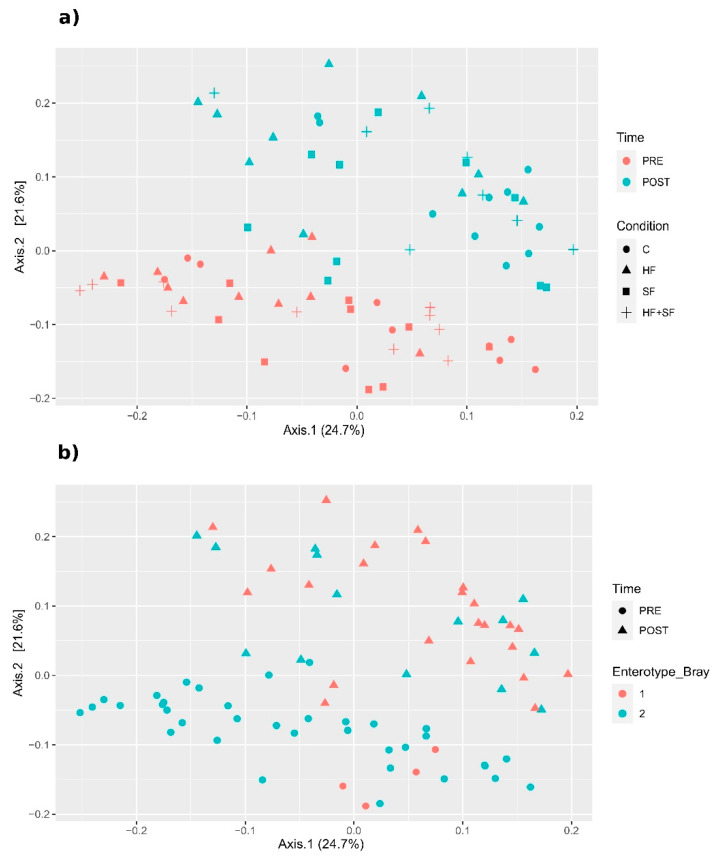
Stratification of the samples. MDS plots based on Bray–Curtis dissimilarity. (**a**) The samples are colored according to the *Time* and shaped according to *Condition* variable (**b**) The samples are colored according to the *Enterotype* variable calculated according to the Bray–Curtis dissimilarity and shaped according to the *Time* variable.

**Figure 3 microorganisms-09-00641-f003:**
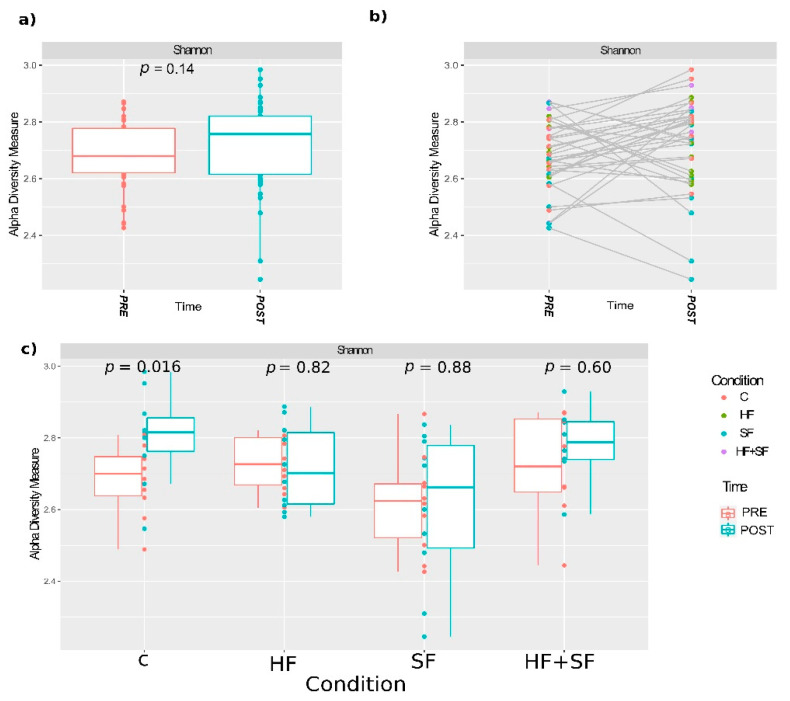
Shannon alpha diversity measure representation for the paired samples. (**a**) Shannon index according to the *Time* variable. Wilcoxon test *p* value is represented; (**b**) variation of Shannon diversity indexes before and after the experiment in each individual mouse. Samples are colored according to the experimental condition. (**c**) Shannon index according to the *Condition* variable (C: controls; HF: heart failure; SF: sleep fragmentation; HF + SF: heart failure and sleep fragmentation. Kruskal-Wallis test *p* values are represented.

**Figure 4 microorganisms-09-00641-f004:**
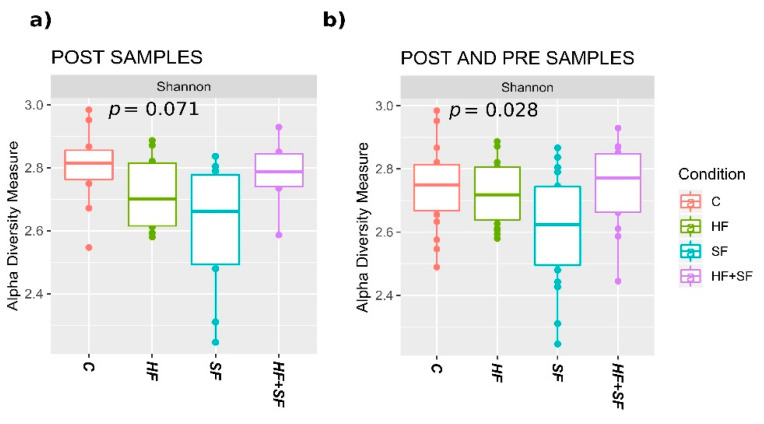
Shannon index representation of the paired samples according to the *Condition* variable. The line inside the boxplot represents the median for each of the groups. (**a**) Considering only post samples. Kruskal-Wallis test showed a non-significant result (*p* = 0.071). (**b**) considering both pre and post samples. Kruskal–Wallis test showed significance (*p* = 0.028).

**Figure 5 microorganisms-09-00641-f005:**
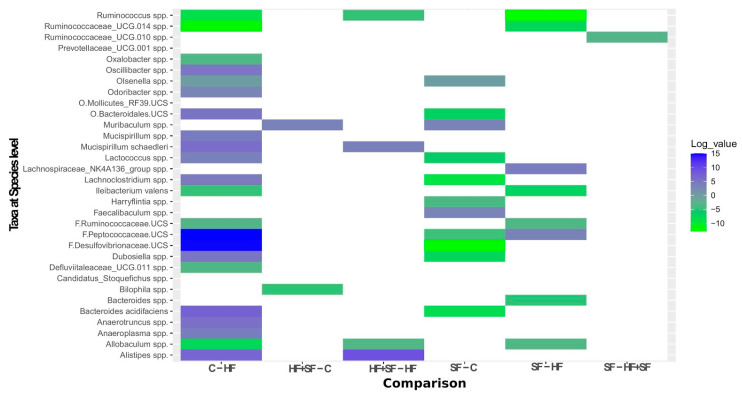
Heatmap representing the 32 significantly differentially abundant taxa at the species level between groups in post samples. The logarithm of only the significant *p*-values is reported (*p* < 0.05), where the values approaching zero are represented as 2.2 × 10^–16^. The sign of the values was transformed to positive or negative according to the direction of the alteration: positive values for increases in the first group within the comparison and negative values for the decreases. Example: A value of 7.218 for *Bacteroides acidifaciens* when comparing C to HF means that this species is significantly higher in C compared to HF.

**Table 1 microorganisms-09-00641-t001:** Differential abundance analysis findings. (**A**) Linear model including all the samples; Fixed effects: Condition and Time variable. Random effects: Batch DNA extraction and Animal (to indicate a paired analysis). (**B**) Linear model taking into consideration only post samples; Fixed effects: Condition and Change of weight (W.change) variables. Random effect: Batch DNA extraction.

Linar Model—Fixed Effect	Phylum	Class	Order	Family	Genus	Species
(A)—Condition	3	5	5	10	23	26
(A)—Time	4	9	10	19	41	47
(B)—Condition	1	2	4	14	30	32
(B)—W.change	1	1	1	3	9	9

**Table 2 microorganisms-09-00641-t002:** Summary of the *p*-values corresponding to the 32 significantly differentially abundant taxa at species level according to both Condition and Change of weight variables.

	Condition	Change of Weight
*Bacteroides acidifaciens*	0.00015	
*Ileibacterium valens*	0.00113	0.00062
*Mucispirillum schaedleri*	0.00125	0.03626
*Olsenella* spp.	2.79 × 10^–25^	
*Bacteroides* spp.	0.00904	
*Odoribacter* spp.	0.03183	
*Muribaculum* spp.	0.01244	
*Prevotellaceae_UCG.001* spp.	0.03238	
*Alistipes* spp.	3.44 × 10^–5^	
O.Bacteroidales.UCS	0.00117	
*Mucispirillum* spp.	0.00408	
*Lactococcus* spp.	0.00262	
*Defluviitaleaceae_UCG.011* spp.	0.04673	
*Lachnoclostridium* spp.	0.00029	
*Lachnospiraceae_NK4A136_group* spp.	0.00637	
F.Peptococcaceae.UCS	1.57 × 10^–6^	0.00019
*Anaerotruncus* spp.	0.00799	0.02487
*Harryflintia* spp.	0.02105	
*Oscillibacter* spp.	0.01505	
*Ruminococcaceae_UCG.010* spp.	0.04265	
*Ruminococcaceae_UCG.014* spp.	8.72 × 10^–6^	
*Ruminococcus* spp.	3.73 × 10^–6^	0.00302
F.Ruminococcaceae.UCS	0.02286	
*Allobaculum* spp.	0.00087	0.01719
*Candidatus_Stoquefichus* spp.	0.04012	
*Dubosiella* spp.	0.00068	
*Faecalibaculum* spp.	0.03229	
*Bilophila* spp.	0.00968	
F.Desulfovibrionaceae.UCS	1.99 × 10^–7^	
*Oxalobacter* spp.	0.01909	
*Anaeroplasma* spp.	0.03002	
O.Mollicutes_RF39.UCS	0.03361	

## Data Availability

The raw sequence files of this study are available in the NCBI Sequence Read Archive (SRA) under the BioProject ID PRJNA662468.

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
