# Peer review of "A Mouse Model Suggests That Heart Failure and Its Common Comorbidity Sleep Fragmentation Have No Synergistic Impacts on the Gut Microbiome"

_microorganisms, 2021, doi:10.3390/microorganisms9030641_

Round 1
Reviewer 1 Report
This paper presents an interesting topic of the relationship between heart failure and gut microbiome. The study was conducted on mice according to scientific principles. In data analysis, appropriate statistical tests were used. I have no comments on the presentation of the obtained results. I find the results interesting and valuable. In my opinion, the work can be published in its present form.
Author Response
Response: We thank the reviewer for his/her very positive comments.
Reviewer 2 Report
Authors studied dependence between heart failure, sleep fragmentation and gut microbiome. They used 16S metabarcoding to assess fecal microbiome composition in mice. They not confirm the synergism between combined heart failure and sleep fragmentation conditions. Studies are made mostly properly, but there are some limitations, which can affect the obtained results: 1. The studied group of mice is very low, 40 mice divided into four subgroups. 2. Microbiome of human differs from microbiome of mice. 3. Results of sequencing and presented microbiome may depend on, among others mice strain, feed, environmental conditions, type of sequencer, sequencing kit, etc. 4. All mice were male. Giving above into consideration, it is difficult to relate these results to changes in human microbiome. Correct research should be performed in humans on many more than 40 patients. I suggest that these results be treated as preliminary research, which should be indicated in the title and abstract. Moreover, if we have several dozen or several hundred microorganisms in each mouse, unfortunately the number of 10 mice in the subgroup is too small for the results to be statistically correct. The volatility can be very large. Therefore, the Authors should first increase the number of animals tested so that there are several dozen mice in each of the subgroups.Author Response
Response: We thank the reviewer for his/her comments. We agree with the reviewer about the limitations of our study. Some of the limitations mentioned such as the possible variability of the results due to different sequencing machines or food for the mice, would be applicable to any empirical result obtained at any lab. The conditions and techniques used for our experiment were homogeneous for all the samples and are thoroughly described so that other groups could reproduce them. Groups of 10 mice each are very typical in the research of both heart failure and sleep fragmentation models. We acknowledge that increasing the number of animals could improve statistical power. However, we want to note that all statistical tests do consider the sample size, so conclusions raised when significance is obtained are still valid. This relates to our choice of only males to reduce the variability, we have a section in the discussion addressing this limitation. Although mice microbiota certainly differ from those of humans, mice are a standard experimental model for many human diseases, as they allow us to perform controlled experiments that would be otherwise impossible in humans. We agree that the preliminary results obtained in mice should be later confirmed by comparing human cohorts affected with either heart failure and sleep fragmentation, or by the two comorbidities. We have now discussed all these limitations, in particular by stressing that this is a model study that should be considered as a preliminary step looking for future patient studies. The only suggestion of the reviewer that we cannot follow is to increase the number of animals. This would be at present an unaffordable task given the difficulty of increasing the number of experimental devices to apply sleep fragmentation and the duration of the challenge. As suggested we underscore the preliminary nature of our findings in the abstract. We have modified the title to make it clear that our results are based on a mouse model.
Reviewer 3 Report
This is a very interesting study. The presentation is very clear and ensures a high readability. The results are well presented and the conclusions are well drawn.
I have no comments and remarks.
Author Response

(The authors gave the same response as above.)

Round 2
Reviewer 2 Report
Thank you for comments of Authors and corrections. I hope next study will be with bigger groups of animals.